# Relative Positioning in Remote Areas Using a GNSS Dual Frequency Smartphone

**DOI:** 10.3390/s21248354

**Published:** 2021-12-14

**Authors:** Américo Magalhães, Luísa Bastos, Dalmiro Maia, José Alberto Gonçalves

**Affiliations:** 1Astronomical Observatory, Faculty of Sciences, University of Porto, 4430-146 Vila Nova de Gaia, Portugal; dmaia@fc.up.pt; 2Interdisciplinary Centre of Marine and Environmental Research (CIIMAR/CIMAR), Faculty of Sciences, University of Porto, 4450-208 Matosinhos, Portugal; lcbastos@fc.up.pt; 3Department of Geosciences, Environment and Spatial Planning, Faculty of Sciences, University of Porto, 4169-007 Porto, Portugal; jagoncalves@fc.up.pt

**Keywords:** smartphone, GNSS, Galileo, dual frequency, precise positioning, long baseline

## Abstract

The use of GPS positioning and navigation capabilities in mobile phones is present in our daily lives for more than a decade, but never with the centimeter level of precision that can actually be reached with several of the most recent smartphones. The introduction of the new GNSS systems (Global Navigation Satellite Systems), the European system Galileo, is opening new horizons in a wide range of areas that rely on precise georeferencing, namely the mass market smartphones apps. The constant growth of this market has brought new devices with innovative capabilities in hardware and software. The introduction of the Android 7 by Google, allowing access to the GNSS raw code and phase measurements, and the arrival of the new chip from Broadcom BCM47755 providing dual frequency in some smartphones came to revolutionize the positioning performance of these devices as never seen before. The Xiaomi Mi8 was the first smartphone to combine those features, and it is the device used in this work. It is well known that it is possible to obtain centimeter accuracy with this kind of device in relative static positioning mode with distances to a reference station up to a few tens of kilometers, which we also confirm in this paper. However, the main purpose of this work is to show that we can also get good positioning accuracy using long baselines. We used the ability of the Xiaomi Mi8 to get dual frequency code and phase raw measurements from the Galileo and GPS systems, to do relative static positioning in post-processing mode using wide baselines, of more than 100 km, to perform precise surveys. The results obtained were quite interesting with RMSE below 30 cm, showing that this type of smartphone can be easily used as a low-cost device, for georeferencing and mapping applications. This can be quite useful in remote areas where the CORS networks are not dense or even not available.

## 1. Introduction

The use of Global Navigation Satellite Systems (GNSS) is nowadays an indispensable tool in areas such as terrestrial, maritime, or aerial transportation, agriculture and forestry, fisheries, financial services, emergency, leisure and recreational activities, mobile services, location, cartography, and many other in which the use of georeferenced information allows for efficiency gains. Providing Positioning, Navigation, and Timing (PNT) services, GNSS systems have been asserting themselves as enhancers of a wide range of applications with an impact on people’s quality of life. These capabilities in mobile phones have evolved in a way that they allow today’s precisions in positioning never achieved in this market segment and, until recently, only achieved using the expensive and high performance GNSS geodetic receivers. Pesyna et al. [1] 2014, demonstrated for the first time that accurate positioning at the centimeter level, was possible based on raw code and phase data from a smartphone-quality GNSS antenna. This possibility can be explored in some areas as a lower-cost alternative to the expensive professional GNSS geodetic equipment.

Three scenarios contributed to this evolution: the first was the introduction of the Android 7.0 Nougat, by the Google company in May 2016 [2], in the smartphones which allow access to the raw code and phase GNSS measurements; the second was the availability of Galileo signals in the smartphones, allowing access to the capabilities of this system characterized by the advanced atomic clocks, the new techniques of signal modulation as the AltBOC, the higher resistance to spoofing and jamming interferences; the third was the introduction, in May 2018, of the Broadcom BCM47755 chip with dual frequency L1/L5 [3] for smartphones, allowing the mitigation of the ionosphere effects, which is one of the biggest sources of error for the L1 only frequency receivers. Furthermore, the L5 frequency allows identifying the signal that makes the direct path (without reflections) easier, minimizing the multipath associated errors, and improving the positioning performance in urban areas.

Several works have been done that show the potential of GNSS, including Galileo-enabled smartphones to reach sub-meter and centimeter accuracy in kinematic or static modes. The authors, Wang et al. [4], used a PPP (Precise Point Positioning) approach to obtain results with a root mean square (RMS) in static mode of 0.20 m in the east, north, and up (ENU) components for the converged solutions and between 0.50 m and 1.09 m in the ENU components for the kinematic test using the dual frequency E1/E5a, L1/L5 from the Galileo, GPS systems and the G1, B1 single frequency from the GLONASS and BeiDou. Critchley-Marrows et al. (2020) [5] also showed a PPP approach with accuracies of 0.33 m in the horizontal and 0.67 m in the vertical components achieved with a dual frequency smartphone, however, a long observation time is needed for the convergence of the solution, which can make the procedure inefficient. According to the authors, real-time PPP is yet to be realized. For many applications, the GNSS post-processing approach has some advantages fundamentally by allowing the edition of the processing parameters, choosing the best combination of the different GNSS systems, choosing atmospheric models and elevations masks, using precise ephemeris and clock information, discarding satellites that disturb the results, usually allowing to reach higher accuracies than the PPP. Paziewski et al. (2021) [6] presented static mode results for GPS L1 solutions in relative positioning, using a very small baseline, with standard deviation values around 40 cm in the float solutions and below 10 cm in the fixed solutions. Static GNSS solutions with an accuracy of 0.10 m from a dual frequency smartphone using a small baseline were obtained by Yan et al. [7]. The same author showed in a kinematic approach accuracy reaching 20 cm with dual frequency and 50 cm with a single frequency smartphone. Uradziński & Bakuła (2020) [8] showed, using a dual frequency smartphone, accuracies of 5 cm for GPS L1 solutions using a reference station a few kilometers away with a survey time of around one hour. Other authors developed works with smartphones with other proposes, Robustelli et al. [9] had done experiments to assess the observations quality and performance of GNSS standalone positioning with dual frequency smartphones, while Liu et al. [10] demonstrated the impact of real-time regional ionospheric correction model can have in the smartphone positioning and Sharma et al. [11] analyzed the antenna limitations in a smartphone-based GNSS positioning approaches. The authors Zangenehnejad & Gao [12] recently published a work about the opportunities and perspectives that the use of GNSS smartphones can be employed in different types of applications in the areas of mapping, navigation, cadastral, and in a wide range of works with georeferencing needs. Some of these works present results of differential positioning using very short baselines and, in general, are based on single frequency measurements, in our experiments we want to assess a long baseline survey methodology with the dual frequency.

In this work, we want to show that dual frequency smartphones can be used as low-cost GNSS surveying tools for relative phase positioning in remote areas, with decimeter accuracy even with distances to a reference station of several hundred kilometers. This is of interest in some areas of the world, such as Africa, South America, or the Antarctic, where the networks of reference stations are sparse and using some of the common GNSS positioning techniques may be difficult, namely the real-time positioning mode (RTK), which relies on short baselines between the rover and the reference station and requires the use of more expensive high-grade geodetic equipment. 

Looking to these challenges and new opportunities we demonstrate in these work that, even using long distance reference stations, positioning with accuracies better than 30 cm is possible using a smartphone. As could be expected, with the increase of the baseline length the GNSS solutions, obtained from relative positioning, show a decrease in quality. A way to mitigate that is to use dual frequency observations provided by some smartphones, namely the Xiaomi Mi8. This device allows to record Galileo and GPS dual frequency (E1/L1 and E5a/L5), code, and phase measurements, which can then be explored for quick ambiguity resolution, cycle slips correction, and minimization of ionospheric errors. As we rely on the use of dual frequency measurements, this work only focuses on the use of the Galileo and GPS systems, discarding the GLONASS and BeiDou because there were no smartphones in the market recording double frequency GLONASS or BeiDou phase measurements. 

This paper is structured in five sections. After this introduction, the experimental setup adopted is described in Section 2 and the signal analyses in Section 3, while Section 4 shows the results obtained and Section 5 present the conclusions.

## 2. Materials and Methods

### 2.1. Campaigns and Equipment Setup

The experimental works, divided into six tests performed in February, March, and November 2021, took place on the facilities of the Astronomical Observatory of the University of Porto (AOUP), using as control points the pillars from a North-South oriented calibration base (Figure 1) with known precise coordinates in ETRS89. The devices used in the experiments were centered at the top of the pillars using a leveling base. The smartphone was positioned so that the GNSS antenna, located in the upper left corner of the screen, was centered over the point. The smartphone available for the experiments was the Xiaomi Mi8 which, at the time of its acquisition, in 2018, was one of the few that allowed access to raw dual frequency data.

For assessing the quality of the smartphone’s measurements two high-grade geodetic receivers were also used in the tests: a Trimble Alloy equipped with a Zephyr Geodetic II antenna and a Septentrio PolaRx5 equipped with PolaNt-MF antenna. The differences between these high-grade devices and low-grade smartphones are several, starting with the mobile device only allowing dual frequency measurements, while geodetic receivers allow multi frequency measurements. The smartphones use a linear polarization antenna type, the PIFA (Planar Inverted F Antenna), with a not controlled radiation pattern, which has a narrow bandwidth, low sensitivity, low multipath suppression, and minimal interference rejection. These antennas are passive and without a signal amplifier due to the critical power consumption in a smartphone. In response to this difficulty, the manufacturers implemented the duty-cycle mechanism to save power, but with impact in the GNSS measurements, making it impossible to acquire continuous carrier-phase measurements. Therefore, in the experiments performed, the duty-cycle off option, available on the Xiaomi, was used to switch off that mechanism. 

The duty cycle does not exist in the geodetic receivers, and they use mostly active antennas with a signal amplifier. The polarization is RHCP (Right Hand Circular Polarization), and the radiation pattern coverage is controlled. The bandwidth in these antennas is wide and they have good sensitivity, as well as a high multipath suppression and good interference rejection. The phase center in the geodetic antennas is calibrated and known in the Antenna Exchange Format (ANTEX), playing a very important role in achieving millimeter accuracy, while for the smartphone antennas no calibration parameters are present in the ANTEX files. The receivers also have a more efficient front-end than the smartphone; this mechanism filters the signals from the antenna reducing the noise and rejecting signals out of the GNSS bands, and also amplifies the signals (ESA (2018) [13]). 

For reference stations, four Continuously Operating Reference Stations (CORS) from the EUREF Permanent GNSS Network [14] were chosen (see Table 1 and Figure 2) in order to have different baseline lengths. Distances vary between 0.2 and 645 km, respectively, for stations Gaia and CEUTA. The GNSS observations from these reference stations were obtained at 1 Hz using the BKG Ntrip Client (BNC) [15] software.

### 2.2. Processing Options

It is known that the GNSS signals are affected by atmospheric delays, ionosphere, and troposphere, that can be minimized or even eliminated, through the application of relative positioning techniques in the case of short baselines, because the receivers are subject to similar effects. With the increase of the baseline distance, the accuracy in the differential solutions decreases basically by 1 ppm × distance. In our case, positioning with the VIGO station, at 121 km, can increase the error by 12 cm, while with CASC and CEUT stations, respectively at 277 and 645 km, may have errors in the order of 28 cm and 65 cm, respectively. 

Processing dual frequency raw observations, using the new signals E5a and L5, from Galileo and GPS, can improve ambiguity resolution for long baselines (ESA (2018) [16]) and reduce the survey time when compared with the use of single frequency observations. In our processing, we used the Double-Difference (DD) model. 

The planning of the observations was done using the GNSS Planning online tool from Trimble [17], in order to choose the best Galileo constellation periods. The smartphones used can provide dual frequency E1/L1 and E5a/L5 measurements from Galileo and GPS signals, and single frequency G1/C1 from GLONASS and BeiDou signals. As our objective was to perform relative positioning with dual frequency measurements using distant reference stations (long baselines), only Galileo and GPS signals were considered in the solution processing. A Septentrio PolaRx5 geodetic receiver, with a PolaNt-MF antenna, was also used as a rover for the comparison of signals and solutions obtained. For the objectives of this work, during 2021, three tests were performed.

To collect the smartphone’s GNSS observations at a rate of 1Hz, the Geo++ RINEX Logger [18] was used, which can generate RINEX (Receiver Independent Exchange Format) files, version 3.03. Due to the fact that the Galileo constellation is still incomplete, and that the GPS constellation has only half of the satellites with the L5 signal, the best approach to have a higher number of satellites with dual frequency signals is the combination of these two systems, which provides also a better PDOP (Position Dilution of Precision). The 1 Hz raw data from the PolaRx5 was collected in SBF file format, which was converted to RINEX 3.04 files using Septentrio RxTools’ SBF Converter [19].

For the processing and analysis of the data the RTKLIB Demo5 software [20], based on the RTKLib [21] software, was used. The analysis of SNR (Signal-to-Noise Ratio), satellite visibility, and geometry was performed using RTKPLOT tools, while Galileo and GPS solutions in a differential mode were computed with RTKPOST. The following processing settings were used to obtain the smartphone static solutions: dual frequency (E1/L1, E5a/L5); elevation mask 15°; broadcast ephemerides; Klobuchar [22] and Saastamoinen [23] models for ionosphere and tropospheric corrections; Antenna parameters from the ANTEX file ngs14.atx [24]. The same settings were used to compute the Septentrio PolaRx5 solutions, where, however, the triple frequency (E1/L1, E5a/L5, E5b/L2) measurements were used.

Six research tests were performed with the objective of acquiring GNSS data to evaluate the positioning performance of a mobile device. To do this, different observation time intervals were carried out at different hours on different days using four reference stations with different baseline lengths to support the analysis of results. This variety of approaches performed aimed to support more reliability in the results, thus, tests were performed to analyze the comparison between smartphones and geodetic receivers. In each survey, smartphones were positioned using a platform mounted on a level base with forced centering (Figure 3), positioned with its GNSS antenna, located in the upper left corner, above the reference points (top of the pillars), to minimize horizontal errors due to displacements. The smartphone’s duty cycle was turned off before starting data acquisition.

Test 1 was done using two identical Mi8 smartphones, designated as smartphone A and smartphone B. The reason for using two devices of the same model was because the two smartphones had been acquired on different occasions and we wanted to assess a possible influence of the hardware and firmware versions on the measurements. Each survey had a duration of 10 min, with the two smartphones 28.5 cm above the pillar, Figure 3a. In order to assess possible disturbances on the GNSS signals, Wi-Fi, Bluetooth, and GSM were turned on and off (flight mode) during the signals’ acquisitions. To perform these analyses, five points on pillars 2, 3, 5, 9, and 11 (see Figure 1) were surveyed in two rounds, simultaneously, with the two devices. In the first round the smartphone A was with flight mode ON and smartphone B with flight mode OFF, in the second round the roles were reversed. The survey started at 10:00 pm on 12 February and had a total duration of about two hours and a half, with a rapid-static survey, 10 min at each point. This test had a small time of survey to have the characteristics of a usual rapid-static survey. 

In Test 2, only smartphone A was used, placed vertically with an antenna offset by 42 cm in relation to the top of pillar 9 (Figure 3b). The survey lasted 2 h, starting at 3:27 pm on 25 February, with the aim of verifying the device’s performance during a longer period of observation. To assess the quality of the signals acquired from the smartphone, measurements from a Trimble Alloy receiver (reference station), with a permanent Zephyr Geodetic II antenna, mounted on a nearby pillar, were used as a reference. The distance between the pillars is about 15 m.

Test 3 was carried out on 13 March, between 9:30 pm and 11:30 pm, surveying points on pillars 5, 9, and 11 for 30 min each with just one smartphone. The device was placed vertically with a displacement height of 30 cm to the pillar reference points. A Septentrio PolaNt-MF antenna with a PolaRx5 receiver was also mounted in the same platform in eccentric mode, 18 cm North of the reference point and with a 20 cm offset height to the antenna reference point (ARP), Figure 3c. The PolaRx5 receiver was used to compare the positioning accuracy on the surveyed pillars between high-grade equipment and the smartphone. As the Alloy receiver is a permanent reference station, it cannot be used to survey other pillars. 

Tests 4, 5, and 6 were carried out with the aim of obtaining measurements during longer periods of observation. The total observation time covered by these tests was 12 h, divided by intervals of 4, 3, and 5 h, respectively, allowing for greater robustness of the results presented in Section 4. These tests were carried out on different pillars, at different times and days, using the same smartphone, which was placed vertically with an antenna displaced 29 cm above the pillar top.

Test 4, held at 4:30 pm on 18 November, had a survey lasting 4 h in pillar 9 (Figure 3d). Test 5 lasted 3 h, starting at 6:30 pm on November 19 in pillar 5 (Figure 3e). Test 6 had a survey time of 5 h, in pillar 11 (Figure 3f), starting at 8:15 am on 22 November. In these last three tests, an aluminum plate was used under the smartphone (Figure 3d–f)) to minimize the multipath effects.

## 3. Satellite Visibility and Observed Signals

In this section, we present, for the tests carried out, the observation conditions regarding satellite visibility and signal availability, as well as the signal quality. 

### 3.1. Satellite Visibility

Figure 4 below, shows the skyplots and geometry during each test, as seen by each GNSS receiver used. 

From the skyplots in Figure 4a,b, we see that the satellites tracked by the two smartphones are slightly different. When we look at the comparison with geodetic grade receivers, the differences are more noticeable, with the mobile device tracking fewer satellites, as seen in Figure 4c–f. This is most notorious in the case of the Galileo constellation, as we can see in Figure 5 and Table 2.

In Table 2, we see, for Galileo and GPS constellations, the minima, and maxima number of satellites in view, obtained for observations with a 0° elevation mask, by the smartphones and by the geodetic receivers used in the experiments.

The constellations geometry, defined by the PDOP average, is also shown in the table. Figure 5 shows the satellite number along the observation interval, seen on each test by the different devices, for Galileo and GPS.

Looking at Table 2, Figure 4a,b and Figure 5a, we see a slight difference in the tracked satellites, and geometry, for the smartphones used during Test 1. For smartphone B we have a better PDOP when we only consider the Galileo satellites, while for smartphone A we see a slightly better PDOP when we consider only the GPS satellites. This can be explained by the differences in the number of satellites shown in Figure 5a, as device B observed more Galileo satellites than device A, whereas in the case of GPS the opposite occurred.

In tests 2 to 6, we see that the smartphone tracked a smaller number of satellites, either Galileo or GPS, when compared to the two geodetic receivers, the Trimble Alloy and the Septentrio PolaRx5. This can be explained by the much lower quality of the antenna but also by the smaller number of channels available on non-geodetic devices.

### 3.2. Signal Quality

The SNR is a very relevant parameter to characterize the quality of the acquired measurements, depending on the tracking, antenna, and bandwidth parameters of the GNSS receiver front-end. In the case of smartphones’ low-cost antennas, their linear polarization, as well as the lower quality front-end, results in a low signal gain. However, SNR can also be affected by other factors such as low satellite signals, malicious interference (jamming), or unintentional interference such as signal obstruction or multipath, which in the smartphone has a low suppression. To mitigate the effect of some of these issues, the observations were made in an open space with good sky visibility and a 15° elevation mask was considered in processing.

The analysis of the SNR of all devices used was performed for the Galileo and GPS signals, E1, L1, E5a, and L5, as shown in the plots at Figure 6, Figure 7, Figure 8 and Figure 9, where each color represents a satellite. The statistical parameters average (AVE), standard deviation (STD), minimum (MIN), and maximum (MAX) are presented in Table 3, Table 4 and Table 5. 

The SNR analysis of Test 1, Figure 6 and Figure 7, and Table 3 shows the results obtained for the two smartphones A and B, with flight mode on or off. In Test 2, the SNR analysis shown in Figure 8 and Table 4 was performed between the signals from smartphone A (flight mode activated) and the Trimble Alloy receiver with the Zephyr Geodetic II antenna mounted on a pillar 15 m away from the mobile device located in pillar 9. In Figure 9 and Table 5, we see the Test 3 analysis made between smartphone A and the signals provided by the multi-constellation geodetic receiver, Septentrio PolaRx5 combined with a PolaNt-MF antenna mounted on the same platform as the smartphone (see Figure 3c).

Looking at the signal strength of the two smartphones during Test 1, we see some small discrepancies between the two devices at both frequencies, with the statistical average varying between 0.5 and 2 dBHz. As mentioned before, data acquisition was performed in different modes: flight mode ON or OFF. This does not significantly affect the signal’s strength in the different frequencies. We see that the SNR values for GPS are a little higher than for Galileo, with maximum values in L1 exceeding 40 dBHz and the L5 above 37 dBHz, while the E1 and E5a are below these values. However, we note that Galileo E1 and E5a standard deviations are better in both devices. The averages of the two systems show differences in the order of 3 dBHz for the E5a/L5 bands and 2 dBHz for the E1/L1 band.

By analyzing the signals acquired with the smartphone in Test 2, we found that the SNR on all signals is lower than that of the Alloy receiver. The averages of the E1/L1 bands on the smartphone are around 38 dBHz, while on the geodetic receiver they are 46 dBHz for the E1 signal and 44 dBHz for the L1 signal. In E5a/L5 signals the differences found are greater with the smartphone presenting values in the order of 34 dBHz, while the Alloy presents values above 46 dBHz. Normally, the difference between a smartphone and a geodetic receiver in terms of SNR is around 10 dBHz, according to several studies presented by some authors (see Liu [10], Paziewski et al. [25] and Liu et al. [26]), the values of 8 and 12 dBHz in the E1/L1 and E5a/L5 bands, respectively, obtained by us are in agreement with a reference value. We also see that the smartphone’s minimum values are below 20 dBHz, while the receiver values are above this value. At maximum, the smartphone does not exceed 50 dBHz, while the receiver has managed to reach values of 54 dBHz in the L5 band. In general, smartphone signals have more fluctuations, especially for low satellites.

Just like in Test 2, in this experiment signals from the smartphone show lower SNR, with average values below 40 dBHz, while for the geodetic receiver, the values reach 47 dBHz. The differences are bigger in the E5a/L5 signals, 12–13 dBHz, while in the E1/L1 signals, the differences are around 5 dBHz. In this test, the minimums between the two devices are much closer than in the previous Test 2, with most values are below 20 dBHz, even for the geodetic receiver PolaRx5. The lower values with this geodetic receiver can be explained mainly by the lower performance of the PolaNt-MF used by the Septentrio receiver when compared with the Zephyr II antenna from the Alloy. This rover antenna does not have such high performance as in the case of the Trimble antenna, a piece of very suitable equipment to work as a reference station. We noticed that STD values from the smartphone are better than the PolaRx5.

The SNR values obtained in tests 4, 5, and 6 were identical to those obtained in the other tests performed and, in order not to become too repetitive, we chose not to present them. The analysis of the signals in the different tests shows that for the smartphones the SNR values for E1 /L1 are higher than for E5a/L5 bands. In all cases, SNR values for smartphones are lower than for the geodetic receivers.

## 4. Results

Differential GNSS solutions for the Galileo + GPS combination and for the GPS-only and Galileo-only observations were calculated with the RTKLPOST software, using the different CORS reference stations listed in Table 1. The measurements were processed using the double-difference method as referred to in Section 2 and applying an Extended Kalman Filter (EKF) algorithm to calculate the final solutions. Tables in Section 4.2, Section 4.3, Section 4.4, Section 4.5, Section 4.6 and Section 4.7 show the statistical parameters average (AVE), standard deviation (STD), and RMS of the positioning errors in the east, north, and up (ENU) directions, which were obtained using the known ETRS89 coordinates of the pillars of the Astronomical Observatory indicated in Figure 1. Figure 10, Figure 11, Figure 12, Figure 13, Figure 14, Figure 15, Figure 16, Figure 17 and Figure 18 show the RMS obtained in the tests performed.

### 4.1. Ambiguity Resolution Ratio

Table 6 shows the ratio of solutions with fixed and floating ambiguities. We see that the fixed ratio of solutions decreases with increasing distance from the reference station and also at shorter observation intervals.

In Table 6, we see that the amount of fix solutions is greater on shorter baselines and in combination with longer survey times. In the long baselines, the number of fixed solutions is residual with almost all floating solutions. In Test 5, CASC and VIGO, two long baselines, 277 and 121 km, we found a good fix solutions ratio, which contributed to a long survey time and also a good number of Galileo satellites. (Figure 5e).

### 4.2. Test 1

Figure 10 above presents the RMS for all Galileo+GPS solutions calculated in Test 1, using observations from the two smartphones with each of the reference stations.

From Table 7, we see RMS values between 20 and 22 cm in the three components for the more distant station CEUT (645 km), which is very interesting and promising for this level of equipment used in the rapid survey (10 min of observation). The other two long baselines VIGO (121 km) and CASC (277 km) have practically all values below 20 cm, with components E and N tending to have values around 15 cm. As expected, the best results were obtained using GAIA, a very small baseline, with the N component below 10 cm and the other two components between 12 and 13 cm. In fact, it was expected that even better values could be achieved with this GAIA station, but previous work experience with a fast survey (<10 min) has shown values of this level. This seems to be a usual accuracy for a quick survey using a smartphone, with the results being close to those obtained in the other tests (Figure 14, Figure 15, Figure 16, Figure 17 and Figure 18). 

Figure 11 shows the RMS for the Galileo + GPS solutions calculated with the two smartphones considering the different reference stations, with flight mode turned on and off. Table 8 shows the statistics obtained.

When looking at the impact of flight mode on or off, we notice that in the case of flight mode off, the results are slightly better, with RMS values mostly below 15 cm. When in flight mode, the results are worse, with values reaching 26 cm. Apparently, WIFI or GSM signals turned on (flight mode off), do not interfere in a way that can degrade the quality of the solutions, presenting even better performance in this mode, which allows us to say that smartphones can be used to collect raw GNSS observations in their normal operating mode.

Figure 12 shows the comparisons of the RMS for Galileo + GPS solutions calculated for each one of the smartphones A and the B. In Table 9, we can see the statistics obtained.

When comparing the two smartphones, we found no significant differences, although smartphones A and B do not track the same satellites (see Figure 5). The results are similar, with maximums below 26 cm and with most values below 20 cm. 

In this Test 1, we also compared the positional performance between the Galileo-only and GPS-only solutions. Figure 13 shows the RMS errors obtained for each GNSS system using observations from two pillars (3 and 5). Due to the smaller number of Galileo satellites available (this constellation is not yet finalized), as seen above in Table 2 and Figure 5, most of the time, there were not enough Galileo satellites to process differential solutions at all points, only these results were presented.

We see a positional performance of Galileo similar to GPS, with most errors below 15 cm, even surpassing GPS in some cases, as in the Up component, which has smaller errors than GPS, except for the VIGO station. In general, the positioning results with both GNSS systems alone are also interesting, being similar to the solution obtained with the combination of both. The best results were for the horizontal components from the Galileo solutions relative to the GAIA station, respectively, 6 cm in the East and 2 cm in the North, while the altitude component Up had an error of 1 cm with the CASC station. Looking at the biggest errors, we found 26 cm for the E component in the Galileo solutions using the VIGO station and 25 cm for the Up component of the GPS using the CEUT station. Bearing in mind that the current limitation on the number of satellites in the Galileo constellation will no longer exist in the future, the positioning performance with this system in itself is quite promising.

### 4.3. Test 2 

Figure 14 and Table 10 show the results obtained during Test 2 for the Galileo + GPS combination using smartphone A on pillar 9, over two hours of observation.

The results of Test 2 show a better behavior than those obtained for Test 1, due to the longer duration of observations (2 h), as we see in Tests 4, 5, and 6, also with a long time of observation, to be presented later. The CEUT station shows RMS values below 15 cm, while the CASC and VIGO stations show values below 10 cm and GAIA, the shortest baseline, below 1 cm.

### 4.4. Test 3

Figure 15 shows the RMS results for the Galileo + GPS combination in Test 3, using the smartphone and a high-grade Septentrio PolaRx5 receiver. Three pillars (5, 9, 11) were surveyed with a duration of 30 min by the two devices simultaneously. Table 11 contains a summary of the statistics obtained.

From the results of Test 3, we see that the errors obtained for the smartphone solutions are similar to the other solutions in Tests 1 and 2, although the biggest RMS errors are reaching the 30 cm in the CEUT station, respectively, in the N and U components. Comparing the two devices, we see that the geodetic receiver had a better performance, especially in the two smallest baselines, GAIA and VIGO, with values below 7 cm, while the mobile device shows for these stations values under 12 cm.

### 4.5. Test 4

Figure 16 and Table 12 show the results obtained during Test 4 for the Galileo + GPS combination using smartphone A in pillar 9 (Figure 1), for four hours.

Test 4 results show values at the level of the other tests, very similar to Test 2. The CEUT station shows RMS values below 20 cm, while the other three stations show values below 5 cm.

### 4.6. Test 5

Figure 17 and Table 13 show the results obtained during Test 5 for the Galileo + GPS combination using smartphone A in pillar 5 (Figure 1), for three hours of observation.

Test 5 results show the best performance of all tests done, with values below 5 cm for all the CORS stations considered. A longer time span with more tracked Galileo satellites may explain this (Figure 5e).

### 4.7. Test 6

Figure 18 and Table 14 show the results obtained in Test 6 for the Galileo + GPS combination using smartphone A in pillar 11 (Figure 1), for five hours of observation.

The Test 6 results show accuracies similar to those obtained in the other tests, with values below 20 cm for the CEUT station, while using the other CORS stations we obtained accuracies below 10 cm for most components.

These results are no better than Test 5, despite the longer observation time. This may be due to the smaller number of Galileo satellites tracked by the smartphone, as shown in Figure 5e,f). A drop in the number of Galileo satellites tracked after the first part of the experiment also contributed to this result. Looking at the satellites tracked by the Alloy receiver (see Figure 5f), we notice that the number of Galileo satellites has not decreased as drastically, so the issue is related to the smartphone’s tracking capabilities. 

The tests performed showed consistent results for differential positioning using long baselines, and, as expected, the smartphone positioning performance was worse in the case of the farthest CEUT station, but still with very good values, below 30 cm. We can conclude that using a smartphone with dual frequency measurements we can still get quite good results, revealing the potential of these low-cost GNSS devices for application in differential positioning based on distant reference stations, with a tendency to improve its performance with the completion of the Galileo constellation and the modernization of the GPS constellation.

It is important to understand whether similar performances can be expected elsewhere across the globe. GNSS signals are affected by atmospheric, ionosphere, and troposphere delays, which can be minimized or even eliminated by applying relative positioning techniques in the case of short baselines, as receivers are subject to similar effects. For our work, we expect ionospheric conditions to be the limiting factor for accuracy, in particular, because an ionosphere-free dual frequency combination could not be used for many of the GPS satellites. At southern European latitudes, such as those in this study, under minimal solar conditions, even for baselines of the order of 100 km, maximum positioning errors using the double-difference technique are about 20 cm, and the RMS of about 5 cm (Crocetto (2008) [27]). Those values are consistent with this study. But during times of disturbed ionospheric conditions, during solar maximum even at mid-latitudes maximum errors around 100 cm and RMS of tens of cm can be expected for baselines on the order of 100 km (Crocetto (2008) [27]). Thus, for equatorial regions where the ionosphere is much more disturbed, even during solar minimum, we do not expect baselines hundreds of km long to achieve the accuracy presented in this paper, although we do expect the low-cost mobile phone solution still function at a level close to that provided by much more expensive equipment.

## 5. Conclusions

The differential results of the Galileo + GPS, Galileo-only and GPS-only combination show that it is possible to obtain good accuracy in the differential positioning mode with long baselines, using dual frequency smartphones (E1/L1, E5a/L5) that allow code and phase measurements. This work presents consistent results in the six tests, with an accumulated observation time of 16 h, based on the comparison with the reference coordinates of the pillars. A good performance was also observed in the evaluation with the results of the geodetic receiver. In general, errors are less than 25 cm in most cases for the long baselines, 277 km (CASC) and 645 km (CEUT), and as expected, the best results were obtained with the shorter baselines, with the GAIA (0.2 km) station showing results better than 15 cm. The accuracies obtained in the long-term surveys show the best results, with RMS values, in most cases below 5 cm, with the furthest station, CEUT, presenting values that in the worst cases reached 20 cm. In the case of the comparison between the two smartphone devices, we verified that the behavior is not strictly the same, showing some differences in the tracked satellites and in the signal strength, but the impact on the obtained solutions was similar. The analysis of the impact of the flight mode on (without Wi-Fi and GSM network) or off, on the performance of the GNSS signal, allowed us to conclude that the differences in signals and results are not significant. About the comparison of the signal strength between the Smartphone with the two geodetic receivers, the Trimble Alloy and the Septentrio PolaRx5, we can see that the smartphone does not reach as high values in the two frequencies of the Galileo and GPS systems as the geodetic receivers, which is not a surprise, due to the lower quality of the smartphone antenna compared to the high performance of the geodetic antennas. However, a good standard of signal strength was obtained, with averages between 30–40 dBHz for E1, L1, E5a, and L5 signals, noting that the E1/L1 smartphone signals reached higher values than the E5a/L5 signals. It should be noted that at the moment we are still somewhat conditioned by the restricted number of satellites with the E5a and L5 signals on dual frequency smartphones, due to the Galileo constellation not being complete and the GPS not being fully modernized either. This forced us to make a plan to choose the best periods of the constellations, especially the Galileo constellation, with the objective of having the largest number of satellites in this constellation, because any satellite broadcasts E1/E5a allowing for always dual frequency terms, while in the case of GPS satellites, a satellite does not necessarily have L1/L5. It is hoped that soon, with the completion of the Galileo constellation and the ongoing modernization of the GPS, it will be easier to carry out any survey having a good number of dual frequency satellites.

The performances we found in this work using a smartphone for differential georeferencing with long baseline remote reference stations, show us a low-cost GNSS equipment with the potential to be a quality tool in georeferencing in response to existing location needs, particularly in remote places of Africa, South America, and Antarctica, where access to dense CORS networks and also limited budgets make it difficult to purchase some expensive high-grade GNSS equipment. It has been shown in several works that different smartphones can provide different quality measures. However, this was not the focus of our work, the aim was to prove the concept that a smartphone can be used to obtain vertical and horizontal accuracies better than 30 cm, even when using reference stations with long baselines.

## Figures and Tables

**Figure 1 sensors-21-08354-f001:**
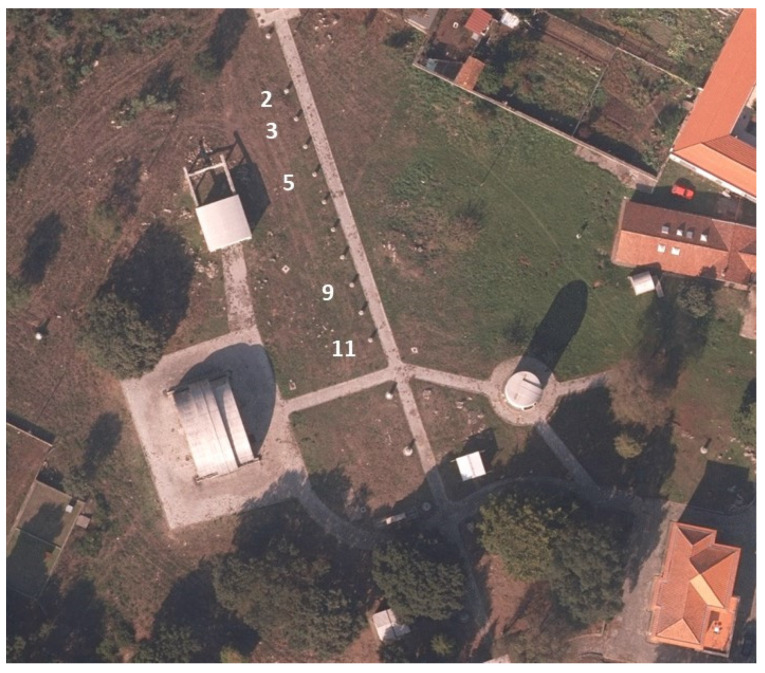
Calibration base at Astronomical Observatory of the University of Porto.

**Figure 2 sensors-21-08354-f002:**
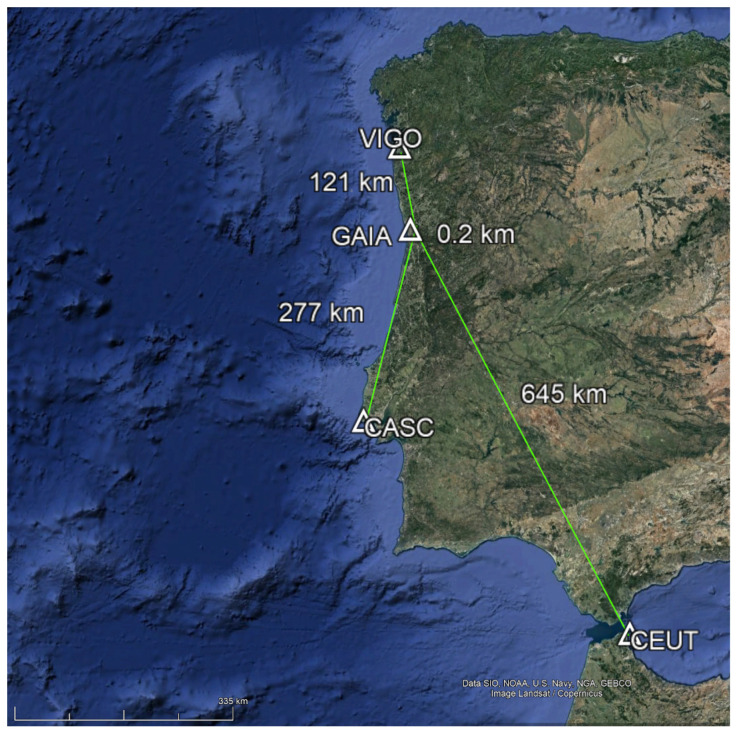
CORS baselines lengths.

**Figure 3 sensors-21-08354-f003:**
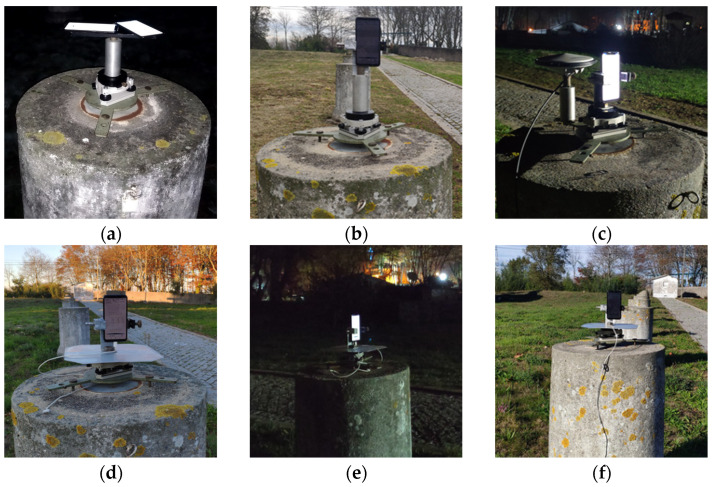
Test 1 with smartphones A and B (**a**); Tests 2 (**b**), 4 (**d**), 5 (**e**) and 6 (**f**) with smartphone A; Test 3 with smartphone A and PolaNt-MF antenna (**c**).

**Figure 4 sensors-21-08354-f004:**
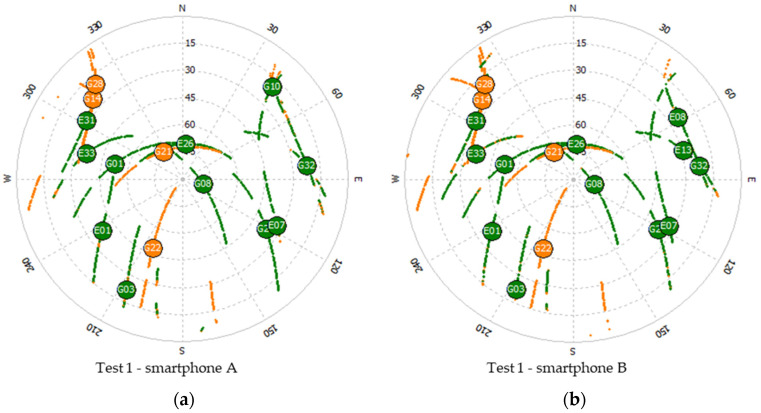
Skyplot: Test 1 smartphone A (**a**) and smartphone B (**b**); Test 2 smartphone (**c**) and Alloy receiver (**d**); Test 3 smartphone (**e**) and PolaRx5 receiver (**f**); Test 4 smartphone (**g**), Test 5 smartphone (**h**); Test 6 smartphone (**i**).

**Figure 5 sensors-21-08354-f005:**
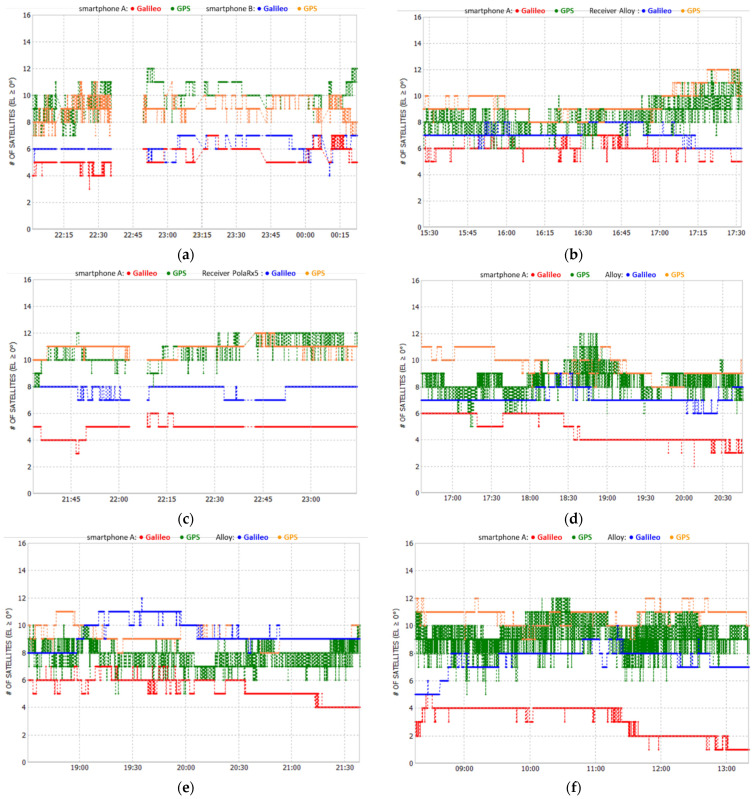
Number satellites for Galileo and GPS: Test 1 (**a**); Test 2 (**b**); Test 3 (**c**); Test 4 (**d**); Test 5 (**e**); Test 6 (**f**).

**Figure 6 sensors-21-08354-f006:**
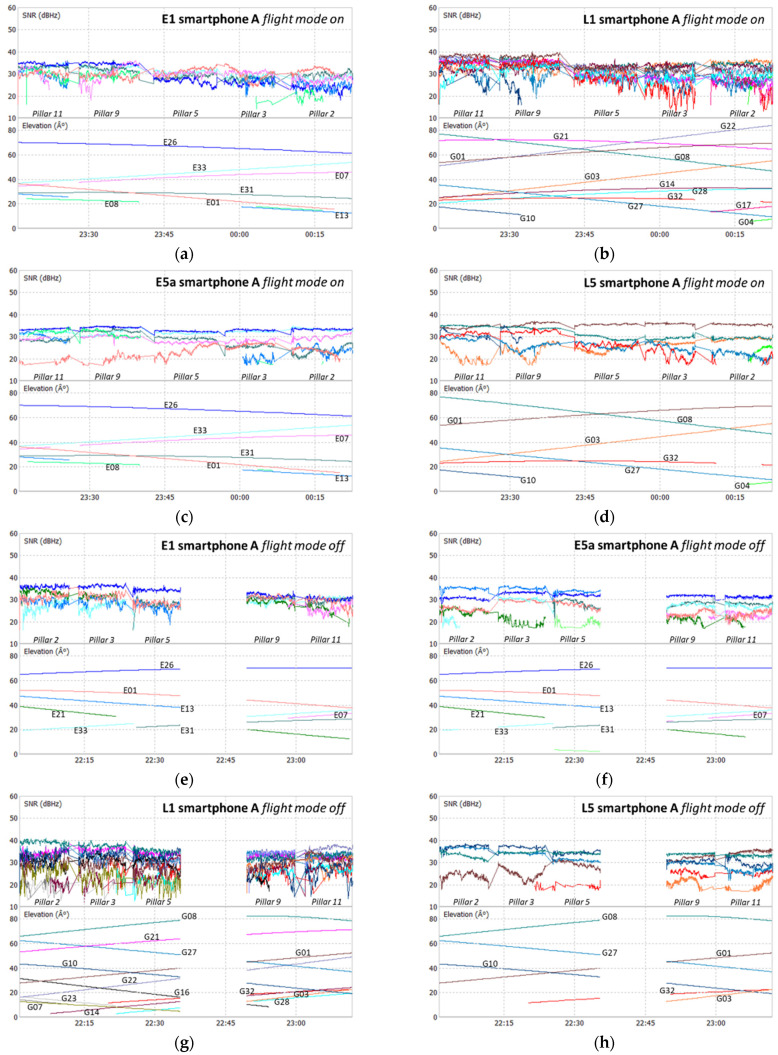
Test 1 SNR smartphone A: E1 flight mode on (**a**); L1 flight mode off (**b**); E5a flight mode on (**c**); L5 flight mode on (**d**); E1 flight mode off (**e**); L1 flight mode off (**f**); E5a flight mode off (**g**); L5 flight mode off (**h**).

**Figure 7 sensors-21-08354-f007:**
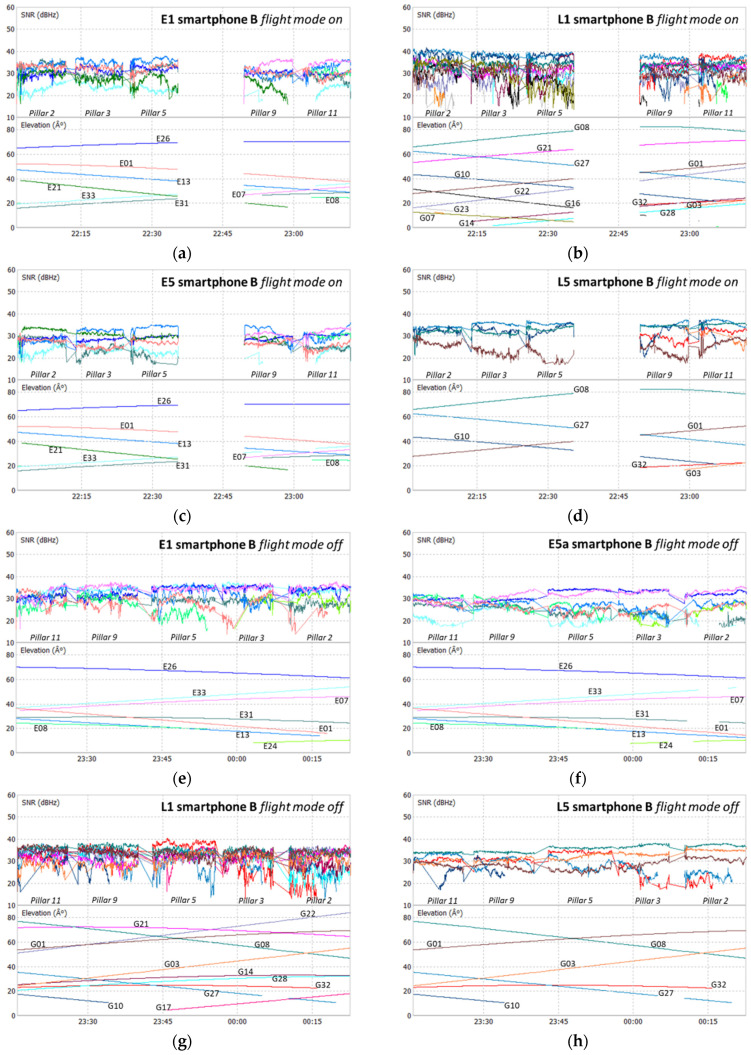
Test 1 SNR smartphone B: E1 flight mode on (**a**); L1 flight mode off (**b**); E5a flight mode on (**c**); L5 flight mode on (**d**); E1 flight mode off (**e**); L1 flight mode off (**f**); E5a flight mode off (**g**); L5 flight mode off (**h**).

**Figure 8 sensors-21-08354-f008:**
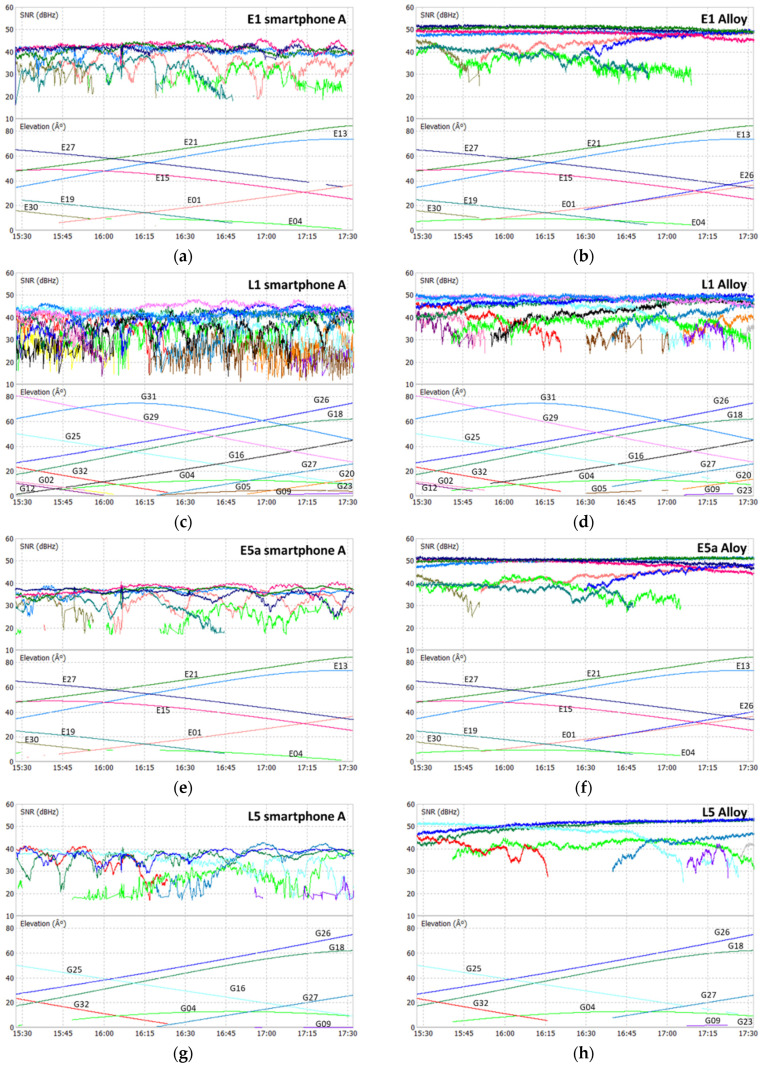
Test 2 SNR: E1 smartphone (**a**) and receiver (**b**); L1 smartphone (**c**) and receiver (**d**); E5a smartphone (**e**) and receiver (**f**); L5 smartphone (**g**) and receiver (**h**).

**Figure 9 sensors-21-08354-f009:**
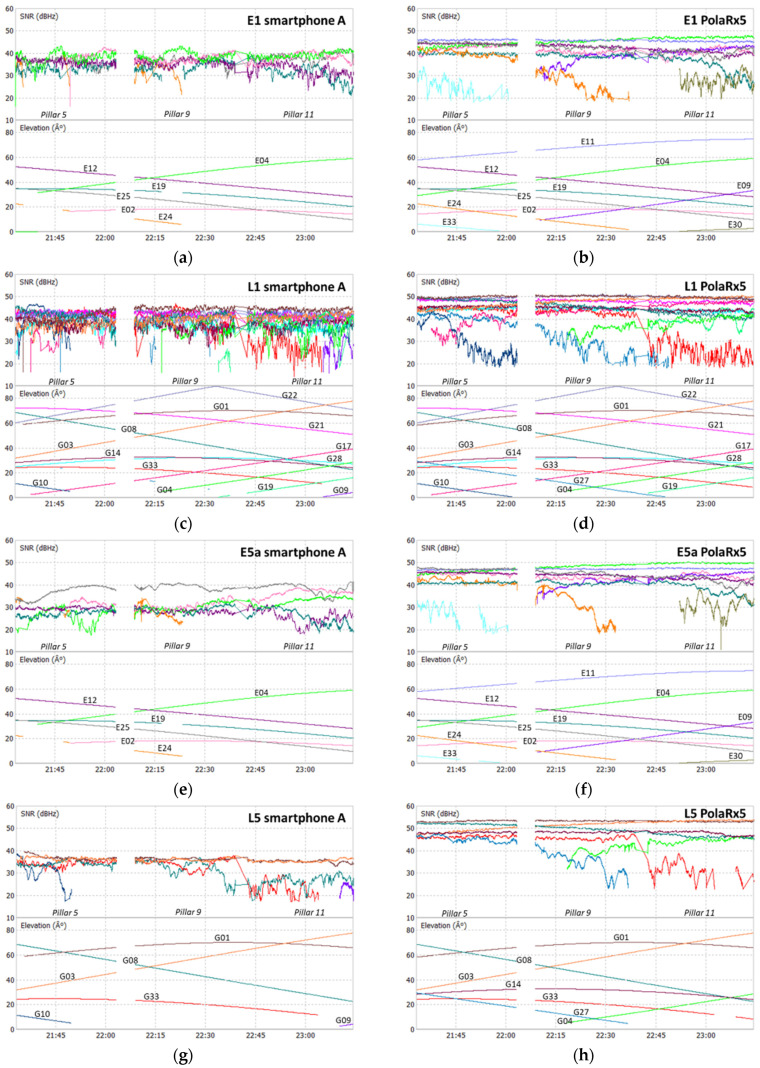
Test 3 SNR: E1 smartphone (**a**) and receiver (**b**); L1 smartphone (**c**) and receiver (**d**); E5a smartphone (**e**) and receiver (**f**); L5 smartphone (**g**) and receiver (**h**).

**Figure 10 sensors-21-08354-f010:**
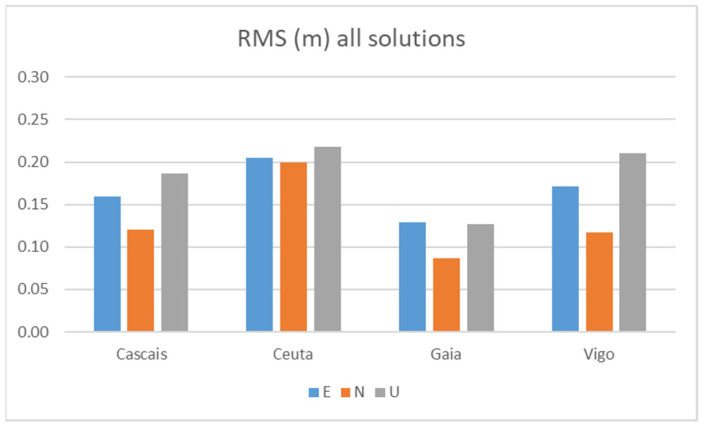
RMS in Test 1 of all Galileo+GPS solutions with the two smartphones.

**Figure 11 sensors-21-08354-f011:**
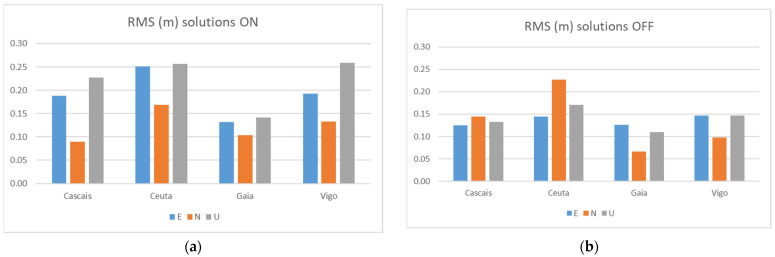
RMS Galileo + GPS solutions from the two smartphones at Test 1: flight mode on (**a**) and flight mode off (**b**).

**Figure 12 sensors-21-08354-f012:**
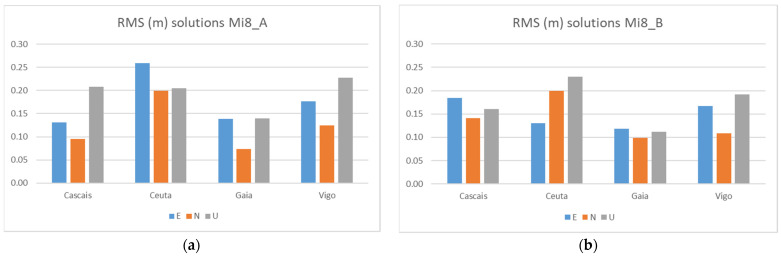
RMS Galileo + GPS solutions from the two Mi8 at Test 1: Mi_A (**a**) and Mi8_B (**b**).

**Figure 13 sensors-21-08354-f013:**
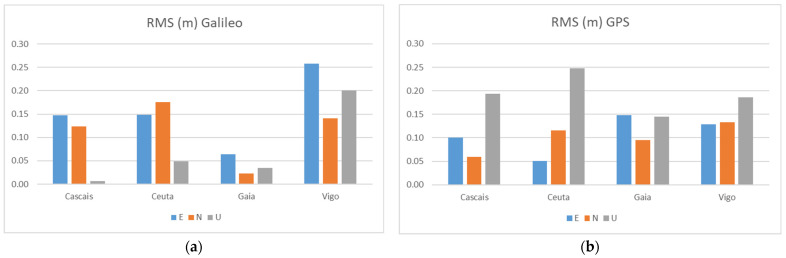
RMS results for the Galileo-only (**a**) and GPS-only (**b**) solutions at Test 1.

**Figure 14 sensors-21-08354-f014:**
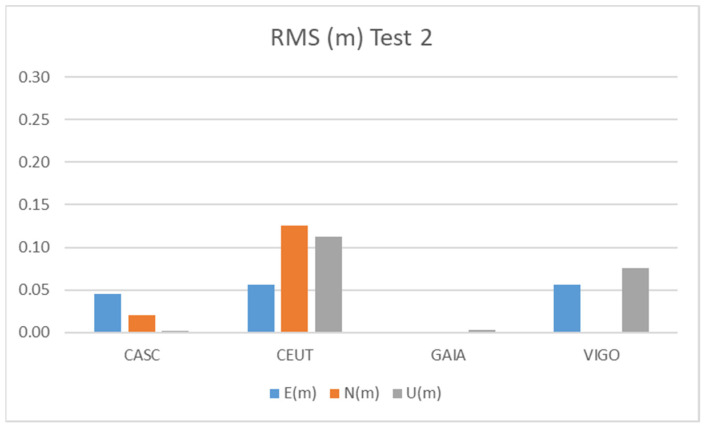
RMS results for Galileo + GPS combination at Test 2.

**Figure 15 sensors-21-08354-f015:**
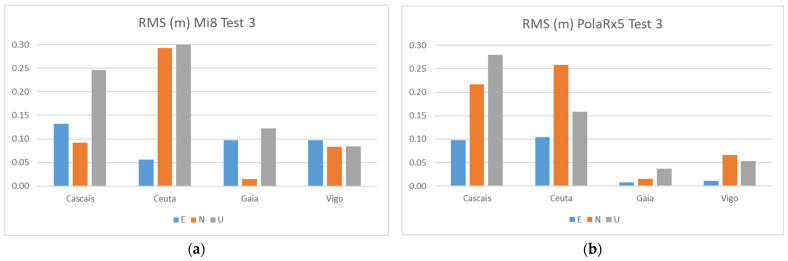
RMS results for Galileo + GPS combination using the smartphone (**a**) and Septentrio PolaRx5 (**b**) at Test 3.

**Figure 16 sensors-21-08354-f016:**
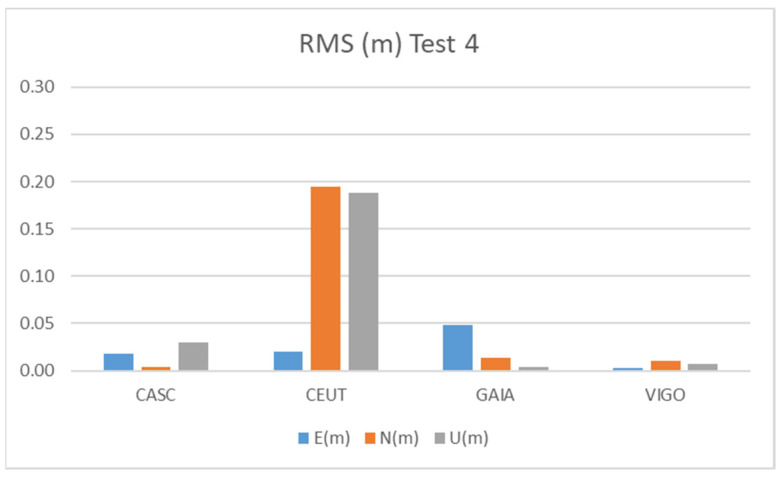
RMS results for Galileo + GPS combination at Test 4.

**Figure 17 sensors-21-08354-f017:**
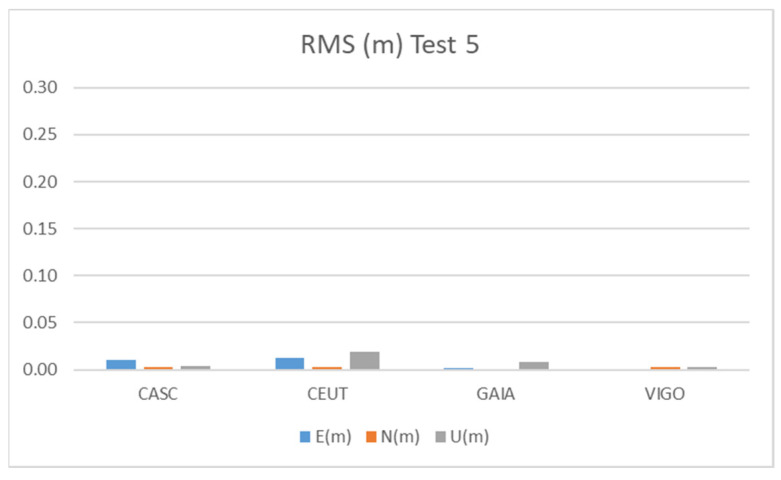
RMS results for Galileo + GPS combination at Test 5.

**Figure 18 sensors-21-08354-f018:**
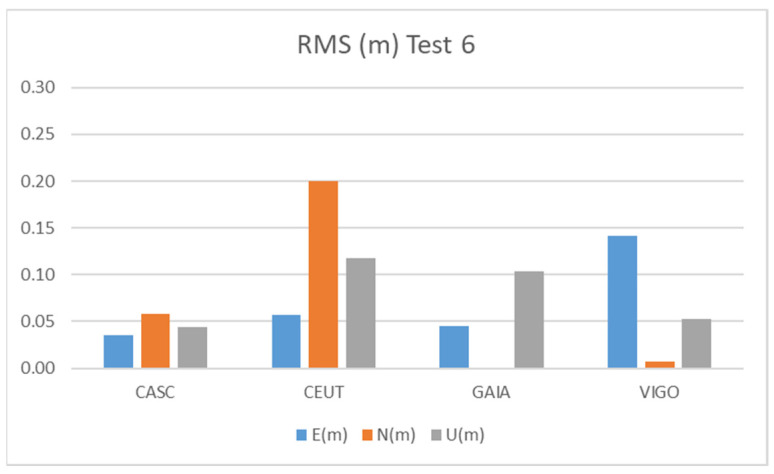
RMS results for Galileo + GPS combination at Test 6.

**Table 1 sensors-21-08354-t001:** CORS from the EUREF used in the surveys.

CORS	Baseline Length (km)	Receiver	Antenna	GNSS Systems	Network	Location
GAIA	0.2	Trimble Alloy	Trimble GNSS-Ti Choke Ring v2	Galileo, GPS GLONASS	RENEP	Gaia Portugal
VIGO	121.1	Trimble NETR9	Trimble L1/L2 Dorne Margoli Choke Ring	Galileo, GPS GLONASS, BeiDou	ERGNSS	Vigo Spain
CASC	277.2	Trimble Alloy	Trimble GNSS-Ti Choke Ring v2	Galileo, GPS GLONASS	RENEP	Cascais Portugal
CEUT	645.5	Trimble NETR9	Trimble L1/L2 Dorne Margoli Choke Ring	Galileo, GPS GLONASS, BeiDou	ERGNSS	Ceuta Spain

**Table 2 sensors-21-08354-t002:** Satellites in view and PDOP for Galileo, GPS, and Galileo + GPS constellations.

	Galileo			GPS			Galileo + GPS			
Test #	# SV Min.	# SV Max.	PDOP	# SV Min.	# SV Max.	PDOP	# SV Min.	# SV Max.	PDOP	Device
1	3	7	3.7	7	12	1.7	11	17	1.4	Smartphone A
	4	7	2.6	7	11	1.8	13	17	1.3	Smartphone B
2	5	7	2.7	6	12	1.6	12	17	1.3	Smartphone
	6	8	2.3	7	12	1.5	13	19	1.1	Alloy
3	3	6	6.1	8	12	1.7	13	17	1.3	Smartphone
	7	8	2.0	10	12	1.4	17	19	1.1	PolaRx5
4	3	6	7.3	6	11	1.7	10	16	1.4	Smartphone
	6	9	2.3	8	11	1.6	15	18	1.2	Alloy
5	4	7	4.1	6	10	1.9	11	16	1.5	Smartphone
	8	11	1.8	8	11	1.6	16	20	1.1	Alloy
6	1	5	4.1	7	12	1.6	9	16	1.4	Smartphone
	5	9	2.3	9	12	1.6	15	20	1.1	Alloy

**#:** number.

**Table 3 sensors-21-08354-t003:** Test 1 SNR statistics for smartphone A and B (units: dbHz).

Flight Mode		Smartphone A				Smartphone B			
		E1	L1	E5a	L5	E1	L1	E5a	L5
ON	AVE	29.6	31.6	29.2	29.2	30.7	32.1	28.0	31.3
	STD	3.1	3.9	4.1	4.8	3.1	4.4	3.5	4.5
	MIN	14.2	12.3	17.0	17.0	15.8	12.9	17.0	17.0
	MAX	36.3	40.2	35.1	37.0	37.9	41.7	36.3	37.7
OFF	AVE	30.2	30.7	28.1	31.0	31.6	32.3	27.6	30.7
	STD	3.0	4.5	4.4	4.8	3.5	3.5	4.0	4.1
	MIN	16.6	12.0	17.0	17.0	13.9	12.2	17.0	17.0
	MAX	37.6	40.9	36.5	38.7	37.7	40.4	35.7	38.4

**Table 4 sensors-21-08354-t004:** Test 2 SNR statistics for smartphone A and Alloy receiver (units: dbHz).

	Smartphone A				Alloy			
	E1	L1	E5a	L5	E1	L1	E5a	L5
AVE	38.6	37.9	34.3	34.8	46.3	44.3	46.3	46.4
STD	4.9	6.3	4.3	5.2	5.1	5.3	5.2	5.6
MIN	17.9	11.4	17.0	17.0	24.8	23.8	24.9	24.7
MAX	46.1	48.3	40.8	42.8	52.6	51.2	52.3	54.2

**Table 5 sensors-21-08354-t005:** Test 3 SNR statistics for smartphone A and PolaRx5 receiver (units: dbHz).

	Smartphone A				PolaRx5			
	E1	L1	E5a	L5	E1	L1	E5a	L5
AVE	36.2	39.6	31.3	33.7	40.8	44.2	43.1	47.3
STD	3.3	3.8	4.9	3.9	5.4	6.1	5.2	5.9
MIN	20.9	14.2	17.3	17.1	18.0	18.0	11.2	22.6
MAX	43.4	46.9	41.8	39.6	48.0	51.3	50.4	54.1

**Table 6 sensors-21-08354-t006:** Ambiguity resolution: Fix and Float ratio of solutions.

	CASC		CEUT		GAIA		VIGO	
	FIX	Float	FIX	Float	FIX	Float	FIX	Float
Test 1	0.4%	99.6%	0.3%	99.7%	1.7%	98.3%	0.1%	99.9%
Test 2	0.9%	99.1%	0.0%	100.0%	89.3%	10.7%	8.3%	91.7%
Test 3	0.0%	100.0%	0.0%	100.0%	78.5%	21.5%	66.8%	33.2%
Test 4	11.0%	89.0%	1.0%	99.0%	83.3%	16.7%	20.3%	79.7%
Test 5	45.7%	54.3%	3.6%	96.4%	57.8%	42.2%	88.9%	11.2%
Test 6	6.4%	93.6%	10.7%	89.3%	15.8%	84.2%	6.8%	93.2%

**Table 7 sensors-21-08354-t007:** Statistic of the errors for all Galileo + GPS solutions in Test 1.

	CASC			CEUT			GAIA			VIGO		
	E	N	U	E	N	U	E	N	U	E	N	U
AVE	−0.024	−0.059	−0.099	−0.076	−0.100	−0.101	−0.007	0.013	−0.044	−0.042	0.077	−0.075
STD	0.162	0.107	0.162	0.195	0.178	0.198	0.132	0.088	0.122	0.171	0.090	0.202
RMS	0.160	0.120	0.186	0.205	0.200	0.218	0.129	0.087	0.126	0.172	0.117	0.211

**Table 8 sensors-21-08354-t008:** Test 1 statistics comparing the flight mode ON & OFF.

		CASC			CEUT			GAIA			VIGO		
Mode		E	N	U	E	N	U	E	N	U	E	N	U
ON	AVE	−0.047	−0.023	−0.172	−0.092	−0.036	−0.151	−0.002	0.030	−0.064	−0.014	0.086	−0.127
	STD	0.192	0.091	0.156	0.246	0.174	0.218	0.138	0.105	0.133	0.203	0.107	0.238
	RMS	0.188	0.089	0.227	0.251	0.168	0.256	0.131	0.104	0.141	0.193	0.133	0.259
OFF	AVE	−0.002	−0.095	−0.026	−0.059	−0.164	−0.050	−0.013	−0.005	−0.024	−0.069	0.068	−0.024
	STD	0.132	0.114	0.137	0.139	0.165	0.172	0.132	0.070	0.113	0.137	0.075	0.153
	RMS	0.125	0.144	0.133	0.145	0.227	0.171	0.126	0.066	0.110	0.147	0.098	0.147

**Table 9 sensors-21-08354-t009:** Test 1 statistics comparing the smartphone A and smartphone B.

		CASC			CEUT			GAIA			VIGO		
Smartphone		E	N	U	E	N	U	E	N	U	E	N	U
A	AVE	−0.093	−0.041	−0.159	−0.190	−0.127	−0.122	−0.070	0.015	−0.076	−0.125	0.084	−0.126
	STD	0.097	0.090	0.142	0.185	0.162	0.174	0.126	0.076	0.123	0.131	0.096	0.199
	RMS	0.131	0.095	0.208	0.259	0.200	0.205	0.138	0.074	0.139	0.176	0.124	0.227
B	AVE	0.044	−0.077	−0.039	0.038	−0.072	−0.080	0.055	0.011	−0.012	0.041	0.069	−0.025
	STD	0.188	0.124	0.164	0.132	0.196	0.227	0.110	0.103	0.118	0.170	0.088	0.201
	RMS	0.184	0.140	0.161	0.131	0.200	0.230	0.118	0.098	0.112	0.167	0.109	0.192

**Table 10 sensors-21-08354-t010:** Statistics results from the Test 2.

	CASC			CEUT			GAIA			VIGO		
	E	N	U	E	N	U	E	N	U	E	N	U
AVE	−0.207	0.144	−0.028	−0.237	0.354	0.334	0.006	0.002	0.053	0.222	−0.019	0.236
STD	0.047	0.005	0.034	0.016	0.013	0.018	0.013	0.008	0.024	0.082	0.018	0.143
RMS	0.045	0.021	0.002	0.056	0.125	0.112	0.000	0.000	0.003	0.056	0.001	0.076

**Table 11 sensors-21-08354-t011:** Smartphone and receiver statistics at Test 3.

		CASC			CEUT			GAIA			VIGO		
		E	N	U	E	N	U	E	N	U	E	N	U
Mi8	AVE	0.087	−0.013	−0.160	0.048	−0.122	−0.059	0.062	0.003	−0.097	0.082	0.072	−0.074
	STD	0.121	0.112	0.229	0.036	0.325	0.362	0.092	0.018	0.091	0.064	0.051	0.049
	RMS	0.132	0.092	0.246	0.057	0.292	0.301	0.097	0.015	0.122	0.097	0.083	0.085
PolaRx5	AVE	−0.047	−0.213	0.244	−0.043	−0.173	−0.053	0.000	−0.003	0.008	−0.009	0.067	0.025
	STD	0.105	0.050	0.169	0.116	0.235	0.182	0.009	0.019	0.044	0.007	0.005	0.057
	RMS	0.098	0.216	0.280	0.104	0.258	0.158	0.007	0.015	0.037	0.011	0.067	0.053

**Table 12 sensors-21-08354-t012:** Statistics results from the Test 4.

	CASC			CEUT			GAIA			VIGO		
	E	N	U	E	N	U	E	N	U	E	N	U
AVE	0.078	−0.033	0.092	0.005	−0.396	−0.392	−0.220	−0.117	−0.058	0.002	−0.083	0.045
STD	0.054	0.026	0.079	0.140	0.194	0.186	0.008	0.005	0.021	0.048	0.054	0.068
RMS	0.018	0.004	0.030	0.020	0.194	0.188	0.048	0.014	0.004	0.002	0.010	0.007

**Table 13 sensors-21-08354-t013:** Statistics results from the Test 5.

	CASC			CEUT			GAIA			VIGO		
	E	N	U	E	N	U	E	N	U	E	N	U
AVE	0.068	0.038	0.042	0.086	0.035	0.133	−0.008	−0.005	0.048	−0.016	−0.040	−0.031
STD	0.091	0.033	0.044	0.030	0.018	0.040	0.026	0.025	0.057	0.014	0.017	0.026
RMS	0.010	0.003	0.004	0.012	0.002	0.019	0.001	0.001	0.008	0.001	0.003	0.002

**Table 14 sensors-21-08354-t014:** Statistics results from the Test 6.

	CASC			CEUT			GAIA			VIGO		
	E	N	U	E	N	U	E	N	U	E	N	U
AVE	0.084	0.240	0.195	−0.117	0.441	0.329	−0.191	0.010	0.296	−0.355	−0.054	0.223
STD	0.168	0.033	0.076	0.209	0.077	0.098	0.083	0.006	0.117	0.123	0.064	0.049
RMS	0.035	0.059	0.044	0.057	0.200	0.118	0.043	0.000	0.101	0.141	0.007	0.052

## Data Availability

Not applicable.

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
