# Peer review of "Relative Positioning in Remote Areas Using a GNSS Dual Frequency Smartphone"

_sensors, 2021, doi:10.3390/s21248354_

Round 1

Reviewer 1 Report

It is of great value to study relative positioning with smartphones. However, the manuscript as a whole lacks novelty. I hope the following comments can help improve the manuscript.

  1. The authors spent a lot of paragraphs on the discussion of the SNR, but the relationship between the SNR and the relative positioning results was not elaborated sufficiently. Did the SNR play a dominant role in relative positioning with smartphones? Are there any other factors affecting the results? Considering the relatively lower SNR with the smartphone data, do we have to adopt some special strategies in data processing? I recommend the authors to consider and answer these questions in the manuscript to make it more evident and valuable.
  2. Only GPS and Galileo measurements were used in the experiment. What about BDS and GLONASS? It would be better if BDS and GLONASS measurements are also analyzed.
  3. Please statement what do you choose Xiaomi Mi8 for your experiment. Reads might have a doubt that the manuscript is promoting the commercial product. To what extent, can the Xiaomi Mi8 represent all kinds of smartphones? Is it necessary to investigate more than one kind of smartphone?
  4. The shortest experiment last only 10min. Even in Test 2, the experiment last only two hours. The limited data can hardly support the conclusion well. To make the analysis more reliable, more data are needed.
  5. As far as I know, there are several other published papers about positioning with smartphone within recent two years. I recommend the authors to study the relevant papers more extensively and include them into the reference list.
  6. Some Figures need to be refined. For example, I guess the different colors in elevation panels of Figure 6 represent different satellites, but in the SNR panels the different satellites are not color-coded. Neither is the other SNR figures. I recommend the authors to unify the figure styles. In addition, it would be better to display some quantitative results in the SNR figures for comparison, such as the average or standard deviation of the SNRs.
  7. There are some typos or grammatical errors, please check all through the paper and polish the language. For example, “Figure 2(a)” in Line 152 should probably be “Figure 3(a)”, “what helps improving” should be “which helps improve” in Line 222, “and the” is repeated in Line 395.

Reviewer 2 Report

The paper investigates the possibility to do long baseline differential GNSS positioning with smartphones. Three slightly different experiments were performed where the raw GNSS observations from phones were collected on well controlled observations points and then evaluated in post processing, using one of four different base stations with varying distances, up to 600 km away.

The reported results show that positioning with a few dm accuracy is feasible with the used smartphones even for the long baseline case and with relatively short observation times of just 10 minutes.

These results are new as I am not aware of another publication that has investigated this processing apporach (DGNSS over long baselines). However, the authors do not sufficiently explain the significant issues of long baselines (unknown atmospheric influences) and what accuracies would therefore be expected theoretically. They also do not explain the differences between the geodetic receiver measurements used as a comparison and the smartphone in any detail.

Consequently, I do not believe that this manuscript is ready for publication as the benefit for the scientific community seems to be limited. It is not sufficient to only report the results, the authors should also explain why the results came out in the way they observed them.

Reviewer 3 Report

The paper presented an analysis for the differential positioning capability of the Xiaomi smartphone. The results show that it indeed have some promising accuracy of positioning. However, all the results were analyzed using off-the-shelf product and open source software (mostly taken from RTKLib). I don't see any significant contribution. The article is more like an advertisement for Xiaomi smartphone.

English writing is confusing. For examples: 1. On Line 13, on what level of precision? 2. On Lines 14~15, there are two "namely" in one single sentence. 3. On Lines 296~299, all of the results are with flight mode "OFF".

There are many such kinds of awkward sentences.

Round 2

Reviewer 1 Report

I appreciate the authors’ hardworking, since they have added some more experiments and improved the manuscript accordingly. However, the manuscript needs to be further improved in at least the following two aspects.

  1. Figures 6-9 repeatedly display the SNR and elevation, making little sense. I did not see any important conclusions drawn from these results. Maybe they can be displayed in a more compact way. Some statistics or comparison probably can make the results more meaningful and readable.
  2. There is still a lot of room for improvement in language, and please check all through the paper and polish the language. The problematic sentences include but are not limited to the followings.

Line 135: The differences between these devices and the much cheaper smartphones are several. Besides, the mobile devices allow only dual-frequency measurements, while those geodetic receivers are multi-frequency.

Line 247: The next more three experiments were similar to the Test 2 and like this the objective was to have a longer observation range, thus having a set of four experiments between the two and five hours of survey, increasing the results assessment consistence presented in section 4.

Line 184: For our tests with were interested in relative positioning with dual-frequency measurements using reference stations far away from the roves (long baselines),

Author Response

Dear Dr./ Mr./Ms. Reviewer 1

We appreciate the time and effort that you have dedicated to providing your valuable feedback and suggestions on our manuscript. Here is a point-by-point response to the reviewers’ comments and concerns.

I appreciate the authors’ hardworking, since they have added some more experiments and improved the manuscript accordingly. However, the manuscript needs to be further improved in at least the following two aspects.

  1. Figures 6-9 repeatedly display the SNR and elevation, making little sense. I did not see any important conclusions drawn from these results. Maybe they can be displayed in a more compact way. Some statistics or comparison probably can make the results more meaningful and readable.
  2. There is still a lot of room for improvement in language, and please check all through the paper and polish the language. The problematic sentences include but are not limited to the followings.

Line 135: The differences between these devices and the much cheaper smartphones are several. Besides, the mobile devices allow only dual-frequency measurements, while those geodetic receivers are multi-frequency.

Line 247: The next more three experiments were similar to the Test 2 and like this the objective was to have a longer observation range, thus having a set of four experiments between the two and five hours of survey, increasing the results assessment consistence presented in section 4.

Line 184: For our tests with were interested in relative positioning with dual-frequency measurements using reference stations far away from the roves (long baselines),

 Three new tables with SNR statistical parameters were add (section 3, page 13-15, lines 328, 343, 362). Figures 6 to 9 have been resized for easier reading.

A review of the indicated sentences below was carried out, as well as a review of the entire manuscript was performed to make it more readable.

We are available to respond to any more questions and comments you might have.

Sincerely,

Américo Magalhães

December 8, 2021

Reviewer 2 Report

The addiditions to the text make the paper more complete. Please consider carefully revisiting the paper especially in terms of readability and clarity of argumentation. It is not so easy to follow your thoughts in the present state of the text.

Author Response

Dear Dr./ Mr./Ms. Reviewer 2

We appreciate the time and effort that you have dedicated to providing your valuable feedback and suggestions on our manuscript. Here is a point-by-point response to the reviewers’ comments and concerns.

The addiditions to the text make the paper more complete. Please consider carefully revisiting the paper especially in terms of readability and clarity of argumentation. It is not so easy to follow your thoughts in the present state of the text.

A review of the entire manuscript was carried out to make it more readable.

We are available to respond to any more questions and comments you might have.

Sincerely,

Américo Magalhães

December 8, 2021

Reviewer 3 Report

No further comments.

Author Response

Dear Dr./ Mr./Ms. Reviewer 3

We appreciate the time and effort that you have dedicated to providing your valuable feedback and suggestions on our manuscript. Here is a point-by-point response to the reviewers’ comments and concerns.

Sincerely,

Américo Magalhães

December 8, 2021